

# Forward Modeling of Spaceborne Active Radar Observations

Isaac Moradi[1,2], Satya Kalluri[3], and Yanqiu Zhu[2]

[1]Cooperative Institute for Satellite Earth System Studies (CISESS)/Earth System Science Interdisciplinary Center (ESSIC), University of Maryland, College Park, Maryland 20741, USA.
[2]NASA Global Modeling and Assimilation Office (GMAO), Greenbelt, Maryland 20771, USA.
[3]Office of Low Earth Orbit Observations, NOAA, Greenbelt, Maryland 20771, USA.

**Correspondence:** Isaac Moradi (imoradi@umd.edu)

**Abstract.** Accurate forward models, particularly radiative transfer models, are essential for the assimilation of both passive and active satellite observations in modern data assimilation frameworks. The Community Radiative Transfer Model (CRTM), widely used in the assimilation of satellite data within numerical weather prediction systems, especially in the United States, has recently been expanded to include an active radar module. This study assesses the new module across multiple radar

frequencies using observations from the Earth Clouds, Aerosols and Radiation Explorer Cloud Profiling Radar (EarthCARE CPR), the CloudSat CPR, and the Global Precipitation Measurement Dual-Frequency Precipitation Radar (GPM DPR).

Simulated radar reflectivities were compared with the spaceborne measurements to assess the impacts of hydrometeor profiles, particle size distributions (PSDs), and frozen hydrometeor habits. The results indicate that both PSD selection and particle shape significantly influence the simulated reflectivities, with snow particle habits introducing differences of up to 4 dBZ in

W-band comparisons. For the GPM DPR, reflectivities simulated using the Thompson PSD showed better agreement with observations compared to those using the Abel PSD. The findings highlight the strong sensitivity of forward radar simulations to microphysical assumptions, underscoring their potential to improve the assimilation of spaceborne radar data in NWP models.

## 1   Introduction

Radiative transfer models (RTMs) serve as critical tools in the field of remote sensing and data assimilation, enabling accurate

representations of how electromagnetic radiation interacts with the atmosphere. These models facilitate the interpretation of satellite observations, which is essential for various applications, including weather forecasting, climate monitoring, and environmental assessments (Liou, 2002; Petty, 2006). The fidelity of such models directly influences the quality of meteorological analyses and predictions, highlighting the need for continuous advancements in radiative transfer methodologies and evaluation of the results (Geer et al., 2017, 2018).

These models are used to translate measurements from spaceborne instruments into quantifiable physical properties of our atmosphere and surface (Saunders et al., 2018; Aumann et al., 2018; Johnson et al., 2023). Within this context, the Community Radiative Transfer Model (CRTM) has emerged as a pivotal collaborative model that enhances both research and operational capabilities in the satellite community. CRTM include capacity to simulate a broad range of spectral channels, from visible to



microwave, which provides a flexible framework for processing and simulating satellite radiances (Chen et al., 2011; Johnson
et al., 2023).

Spaceborne radar missions, such as the Global Precipitation Measurement (GPM) Dual-frequency Precipitation Radar (DPR)
and the Earth Clouds, Aerosols and Radiation Explorer (EarthCare) Cloud Profiling Radar (CPR), provide vertical profiles
of cloud and precipitation structures that cannot be obtained from passive sensors, offering essential constraints for model
development and improving our understanding of key atmospheric processes. Accurate radiative transfer modeling is essential
to properly interpret how radar signals interact with hydrometeors at different frequencies (Prigent, 2010; Battaglia et al.,
2020, 2024). This is required for different purposes such as estimating hydrometeor properties from radar signals or assimilation
of these observations in the NWP models (Battaglia et al., 2024; Moradi et al., 2023).

Currently, several NWP centers use active sensor simulators tailored to their own systems for the assimilation of spaceborne
radar observations. Di Michele et al. (2012) used the ZmVar forward operator to compare ECMWF forecasts with CloudSat
radar data, showing that simulated reflectivities are highly sensitive to assumptions about precipitation fraction, particle size,
and hydrometeor properties.

Okamoto et al. (2016) demonstrated that direct assimilation of GPM DPR reflectivity has the capability to improve the
forecasts. Using an ensemble-based variational scheme with careful preprocessing (quality control and superobbing), DPR
data provided vertically resolved information to better constrain rain mixing ratios and updrafts, while complementing GMI
radiance assimilation. Results showed that combined DPR and GMI assimilation outperformed either dataset alone, with DPR
helping to mitigate biases such as excessive snow and spurious rain increases.

Recent developments in RTTOV-SCATT have extended its capabilities to simulate active radar observations alongside pas-
sive microwave sensors within a consistent radiative transfer framework. The radar simulator introduced in version 13 includes
improvements to melting-layer parameterizations, with a revised scheme shown to better reproduce GPM DPR observations,
particularly below the freezing level. This enhanced capability can be used in either NWP model evaluation or data assimilation
(Geer, 2021; Mangla et al., 2025).

The active sensor module in CRTM is integral for assimilating remote sensing observations specifically from spaceborne
active instruments such as EarthCARE CPR and GPM DPR (Moradi et al., 2023). This module has been successfully integrated
into the Joint Effort for Data assimilation Integration (JEDI) framework, which supports an operational environment that
emphasizes real-time data assimilation. The evaluation of CRTM's active sensor module, particularly for Ku, Ka, and W bands,
will contribute significantly to the readiness of the JEDI system for the assimilation of forthcoming spaceborne observations.

This paper aims to assess the performance of the CRTM active sensor module in simulating radar observations across the
Ku, Ka, and W bands using atmospheric and hydrometeor profiles collocated with radar observations. Section 2 provides an
overview of the CRTM, Section 3 describes the instruments, Section 4 presents the results, and Section 5 offers the conclusions.



## 2 Forward Models in Data Assimilation

Radiative transfer models are the cornerstone of data assimilation frameworks used to incorporate satellite observations from both passive and active instruments into NWP models. These models, often refereed to as the forward model or observation operator, simulate satellite measurements based on atmospheric state variables and hydrometeor profiles provided by the NWP models (Zhou et al., 2023; Liang et al., 2023). Modern data assimilation systems predominantly employ four-dimensional variational (4D-Var) methods. 4D-Var aims to optimally estimate the initial conditions by minimizing a cost function that represents the difference between model predictions and observational data over a specified temporal window, thus ensuring a fit between the model trajectory and incoming data in four-dimensional space (three spatial dimensions plus time) (Rabier et al., 2000).

The general formulation of the data assimilation cost function can be expressed mathematically as follows:

$$J(x) = \frac{1}{2}(\mathbf{y} - \mathbf{H}(x))^T \mathbf{R}^{-1}(\mathbf{y} - \mathbf{H}(x)) + \frac{1}{2}(x - x_b)^T \mathbf{B}^{-1}(x - x_b) \tag{1}$$

where, J(x) is the cost function to be minimized, ($\mathbf{y}$) represents the vector of observations, ($\mathbf{H}(x)$) is the forward model that maps model state to observation space, ($x_b$) symbolizes the background state (typically a forecast), ($\mathbf{B}$) is the background error covariance matrix, and ($\mathbf{R}$) denotes the observational error covariance matrix (Rabier et al., 2000).

### 2.1 Radar forward model

The power received by a radar antenna depends on the transmitted power, target scattering, and atmospheric attenuation. By normalizing the backscattered signal with the transmitted power, the instrument characteristics are removed, yielding the radar reflectivity. Mathematically, the radar equation can be expressed as follows:

$$R = \frac{10^{18}\lambda^4}{\pi^5 |k_w|^2}\beta_b, \ R_a \quad = \frac{10^{18}\lambda^4}{\pi^5 |k_w|^2}\Gamma\beta_b, \tag{2}$$

where $\lambda$ is the radar wavelength (m), $k_w$ is the dielectric factor of liquid water, $\beta_b$ is the volume backscattering coefficient (m$^{-1}$), and $\Gamma$ is the two-way transmittance accounting for atmospheric attenuation. The attenuation is computed from the layer extinction coefficient, accounting for both absorption and scattering by gases and hydrometeors (Moradi et al., 2023). Reflectivity ($R$) and attenuated reflectivity ($R_a$) are typically expressed in logarithmic units in dBZ as follows:

$$R_e = 10\log_{10}(R), \ R_{ea} \quad = 10\log_{10}(R_a), \tag{3}$$

PSDs are used to derive bulk scattering properties from single-particle scattering characteristics. They provide the number density of particles per unit diameter of particle, $n(D)$ in $m^{-3}m^{-1}$, which is used to calculate the volume backscattering coefficient $\beta_b$ as follows:





$$\beta_b = \int\limits_0^\infty \sigma_b(D), n(D), dD, \tag{4}$$

where $\sigma_b$ (m$^2$) is the backscattering cross section of a particle with maximum dimension $D$(m).

In CRTM, $\beta_b$ is expressed using the mass backscattering coefficient $k_b$ (m$^2$ kg$^{-1}$) and the cloud water density $\rho_h$ (kg m$^{-3}$):

$$\beta_b = \rho_h k_b = \frac{\psi_h k_b}{d_x}, \tag{5}$$

where $\psi_h$ (kg m$^{-2}$) is the layer-integrated cloud water content provided as input to CRTM, and $d_x = d_z/\cos(\theta)$ is the slant-path layer thickness, with $d_z$ the vertical layer thickness and $\theta$ the zenith angle. The coefficient $k_b$ is obtained as follows:

$$k_b = \frac{\int \sigma_b(D), n(D), dD}{\int m(D), n(D), dD} = \frac{\int \sigma_b(D), n(D), dD}{\int \rho(D)V(D), n(D), dD}, \tag{6}$$

where $m(D)$ is the particle mass (kg), $V(D) = \pi D^3/6$ is the particle volume, and $\rho(D) = m(D)/V(D)$ is the particle density.

## 2.2 Community Radiative Transfer Model

The Community Radiative Transfer Model (CRTM), developed by the Joint Center for Satellite Data Assimilation (JCSDA), is a computationally efficient and flexible framework for simulating satellite observations across passive microwave, infrared, and visible sensors under both clear-sky and cloudy-sky conditions (Johnson et al., 2023). The CRTM capability has been recently extended with the addition of an active sensor module to facilitate the assimilation of spaceborne radar observations (Moradi et al., 2023)

The model relies on pre-computed lookup tables to represent absorption and scattering processes: absorption coefficients are tailored to each instrument, whereas hydrometeor scattering properties are generalized as functions of particle mass, frequency, and temperature. As a result, a unified set of hydrometeor lookup tables can be applied to all sensors operating within the same spectral domain. In particular, both microwave radiometers and radar systems utilize the same hydrometeor scattering databases due to their shared frequency range. Recent developments have introduced Discrete Dipole Approximation (DDA) lookup tables, extending CRTM's applicability to microwave and radar instruments operating from 10 GHz to 800 GHz (Moradi et al., 2022).

To account for multiple scattering, CRTM integrates two advanced solvers: the Advanced Doubling–Adding (ADA) method (Liu and Weng, 2006) and the Successive-Order-of-Interaction (SOI) method (Heidinger et al., 2006). While these solvers are primarily applied to attenuation computations, backscattering effects are treated using the single-scattering approximation. By default, CRTM employs ADA, which is also adopted in the present study.

Full radiative transfer simulations for microwave and radar instruments in CRTM require atmospheric state variables (temperature, water vapor, and pressure) in conjunction with hydrometeor water content profiles, including snow, hail, graupel,



cloud ice, cloud liquid water, and rain. In the case of radar observations, these inputs are essential for computing the two primary processes governing radar measurements: volume backscattering and atmospheric attenuation. Although simulations may be performed for an isolated hydrometeor species (e.g., only snow), such an approach neglects the scattering and attenuation contributions from other hydrometeors.

## 3    Spaceborne Radar Observations

Although ground-based radars are widely used, particularly for measuring precipitation and extreme weather, only a few spaceborne radars currently exist. However, radar instruments are becoming increasingly common, such as those planned for NASA's Atmosphere Observing System (AOS) (da SIlva et al., 2021).

Radar instruments transmit electromagnetic pulses that travel through the atmosphere, interact with hydrometeors, and scatter a portion of the signal back to the antenna. The strength of this backscattered signal, or reflectivity, depends on the scattering properties of the hydrometeors and the attenuation along the signal path (Moradi et al., 2023).

Accurate atmospheric and hydrometeor profiles are therefore required to evaluate radar forward models using observations from different instruments. However, no single dataset is available that provides input hydrometeor profiles suitable for all the radar instruments evaluated in this study. As a result, a separate retrieval-based hydrometeor dataset was used to evaluate the forward model for each instrument considered in this study.

### 3.1    EarthCARE CPR

The European Earth Clouds, Aerosols and Radiation Explorer (EarthCARE) satellite carries four instruments to study the role of clouds and aerosols in Earth's climate (Illingworth et al., 2015). Among these, the Cloud Profiling Radar (CPR) is a Doppler-capable radar that penetrates clouds and light precipitation, providing detailed measurements of vertical structure, particle size, water content, and velocities.

The CPR is equipped with a 2.5 m antenna deployed shortly after launch. CPR operates at 94 GHz and provides a vertical resolution of 500 m. The high sensitivity of the radar ensures the detection of weak microwave backscatter from hydrometeors, while internal calibration modes maintain the linearity and precision of the signal processing system.

In our study, we used cloud-related microphysical parameters derived from the CPR-ATLID synergy cloud algorithm, as described in Okamoto et al. (2024) and Sato et al. (2025). Specifically, we utilized the updated vBb/v1.1 version of the algorithm,

which provides cloud masks and cloud type classifications while improving detection near the ground by reducing surface clutter contamination compared to previous versions. The dataset includes Level 2 cloud and precipitation microphysics retrievals from EarthCARE's CPR, ATLID lidar, and multispectral imager.

### 3.2    GPM DPR

The Dual-frequency Precipitation Radar (DPR) onboard the NASA-JAXA Global Precipitation Measurement (GPM) Core

Observatory is a flagship instrument for observing precipitation worldwide. The DPR consists of two co-aligned radars: the Ku-



band Precipitation Radar (KuPR) operating at 13.6 GHz and the Ka-band Precipitation Radar (KaPR) at 35.5 GHz. Together, they provide 3-dimensional observations of rainfall and snowfall, enabling accurate estimation of precipitation rates, drop size distributions, and vertical precipitation structure. By comparing differential attenuation between the two frequencies, DPR can distinguish between rain and snow, improving sensitivity to light precipitation and snowfall in mid-latitude regions. The

dual-frequency capability enhances the accuracy of hydrological and meteorological analyses, supports numerical weather prediction, and allows scientists to study storm microphysics in unprecedented detail (Hou et al., 2014).

Each radar has a nadir spatial resolution of approximately 5 km with a range resolution of 250 m, covering a swath of up to 245 km. The high sensitivity and advanced calibration techniques of DPR allow it to detect both weak and intense precipitation over land and ocean, day and night, providing vital data for climate research, hydrological studies, and weather forecasting

(Hou et al., 2014; Skofronick-Jackson et al., 2017).

For our study, we used precipitation and environmental data from Version 07 of the GPM DPR Precipitation Profile L2A dataset (GPM_2ADPR) (GPM Science Team, 2021). This dataset provides single- and dual-frequency-derived precipitation estimates from the Ku- and Ka-band radars onboard the GPM Core Observatory. The products include precipitation retrieved from the wide-swath Ku-band (245 km), the narrow-swath Ka-band (125 km), and dual-frequency measurements over the

narrow swath. For dual-frequency retrievals, the inner swath fields of view of Ku- and Ka-band measurements are co-aligned, allowing derivation of particle size distribution and improved estimation of rainfall rate and equivalent liquid water content (Precipitation Processing System (PPS) At NASA GSFC, 2021; GPM Science Team, 2021).

These DPR retrievals also include environmental profiles assumed in the Level 2 retrieval algorithm, such as atmospheric temperature, pressure, and water vapor. The radar measurements are performed at each range bin along the slant path of the

radar instrument field of view (IFOV) (GPM Science Team, 2021).

### 3.3 CloudSat CPR

CloudSat, launched in 2006 into a sun-synchronous orbit at an altitude of approximately 700 km, was designed to investigate the role of clouds in the climate system. It carried a CPR, which operates at 94.05 GHz with a bandwidth of 0.3125 MHz, providing radar reflectivities with a vertical resolution of 500 m. CPR measurements are averaged onboard over 0.16 s, yielding

a footprint of 1.4 km by 1.7 km. These are further averaged during ground processing to an along-track resolution of 3.5 km (Stephens et al., 2002). In this study, we used radar reflectivity from the CloudSat Level 2B-GEOPROF-R05 dataset (Marchand et al., 2008), which ensures significant radar echoes from hydrometeors rather than noise or clutter.

Cloud liquid and ice water content profiles were obtained from the Release 05 Level 2B-CWC-RVOD dataset, derived from CloudSat radar reflectivity and Aqua MODIS cloud optical depth using the algorithm described in (Leinonen et al., 2016).

Inputs to this algorithm include CloudSat radar reflectivity (Marchand et al., 2008) and MODIS cloud optical depth onboard Aqua (Platnick et al., 2003). Snow water content was obtained from the Level 2C Snow Profile (*2C-SNOW-PROFILE*) dataset (Wood and L'Ecuyer, 2018), retrieved using an optimal estimation technique. Due to ground clutter, the base of the retrieved snow layer may be truncated above the surface (Wood and L'Ecuyer, 2018). While graupel is not explicitly retrieved, it is likely encompassed within the ice water content, and its inclusion has a limited effect on RT simulations. This is because ice



and graupel exhibit similar microwave scattering properties, provided comparable particle shapes are assumed. Hail was not considered in RT simulations due to the absence of hail water content retrievals.

Finally, ERA-Interim atmospheric profiles of temperature, water vapor, and pressure (Dee et al., 2011) were interpolated to CloudSat geolocations and used as input for CRTM. These profiles have been validated against both satellite observations and in-situ radiosonde data (Virman et al., 2021), and are considered sufficiently accurate for calculating gaseous attenuation.

## 4 Results

The results are categorized into three sections: first, evaluating the CRTM forward model for the Ku/Ka and W bands; second, investigating how the particle size distribution impacts reflectivity calculations in the Ku/Ka band; and finally, exploring the impact of the shape of the snow particles on reflectivity computations for the W band.

### 4.1 Evaluation of the forward model

Evaluating a forward radiative transfer model at radar frequencies requires atmospheric variables such as pressure, temperature, and water vapor, in addition to hydrometeor profiles including cloud liquid water, ice, rain, and snow water content. While the DPR frequencies are primarily influenced by rain backscatter, the CPR W-band is more sensitive to backscatter from frozen hydrometeors, though backscatter from low-altitude rain can also be significant. Although liquid cloud droplets are typically too small to contribute directly to backscatter, they affect the measured reflectivity through attenuation.

Multiple datasets, including reanalysis products and numerical model simulations, can reliably provide the atmospheric state variables, such as pressure, temperature, and water vapor, required for RT simulations. However, they face substantial limitations in accurately representing hydrometeor distributions. NWP models, in particular, often struggle to produce cloud fields with sufficient accuracy to be used to validate radiative transfer simulations. Despite this, combined retrievals of hydrometeors constrained by radar observations offer a more reliable basis for forward model evaluation. While these retrievals do involve radiative transfer calculations themselves, the RT model used differs from CRTM; thus, it is justified to use these retrievals, alongside reanalysis-based atmospheric profiles, for the validation of our forward model (Moradi et al., 2023).

We evaluated the forward model using multiple overpasses from CloudSat CPR and GPM DPR during hurricanes, as well as global observations from EarthCARE CPR. The CloudSat CPR data are from August 19, 2023, with over 160,000 valid observations; the EarthCARE CPR data are from the first week of May 2025, with over 12 million valid observations; and the GPM DPR data are from several days in September 2017, with more than 50 million valid global observations. We screened out any observations below the minimum detectable reflectivity thresholds proposed in the literature for these instruments: 15.46 and 19.18 dBZ for GPM DPR Channels 1 and 2 Liao and Meneghini (2022), -30 dBZ for CloudSat CPR Arulraj and Barros (2017), and -35 dBZ for EarthCARE CPR Wehr et al. (2023).

Table 1 summarizes the mean, standard deviation, and selected percentiles of the differences between simulated and observed reflectivities for all three instruments using global observations. The PSDs used in the simulations are summarized in the table.



The PSD for frozen hydrometeors follows Field et al. (2007), while those for rain and liquid clouds are based on either Thompson et al. (2008) or Abel and Boutle (2012).

For EarthCARE CPR, the simulated reflectivities are on average slightly higher than observations by 0.59 dBZ, with a standard deviation of 6.18 dBZ. The percentile distribution shows that 90% of the differences are within 10 dBZ, while the median (P50) difference is 1.03 dBZ, and half of the cases have differences smaller than 5 dBZ.

CloudSat CPR also shows a positive bias, with simulations exceeding observations on average by 2.62 dBZ. The spread is similar to EarthCARE, with a standard deviation of 6.46 dBZ. The median difference is 3.34 dBZ, and 50% of the cases have absolute differences below 9.0 dBZ.

In contrast, GPM DPR shows a negative bias for the Ku-band (channel 1) global data, with an average difference of –2.04 dBZ and a narrower spread (standard deviation 3.39 dBZ). The Ka-band (channel 2) global results are closer to neutral, with a mean difference of –1.46 dBZ and slightly larger variability (Std 3.76 dBZ). Finally, for the DPR Ku and Ka bands results using the Abel 2012 PSD, the biases are strongly negative, at -22.87 dBZ and -22.68 dBZ, respectively, with similar spreads to the results from Thompson 2008 PSD (3.3–4.2 dBZ).

| Inst | Chan | PSD (Ice/Liquid) | Mean | Std | P5 | P25 | P50 | P75 | P95 | #obs |
|------|------|------------------|------|-----|-----|------|------|------|------|------|
| CPR-EarthCARE | 1 | F07/T08 | 0.59 | 6.18 | -9.81 | -3.58 | 1.03 | 4.89 | 10.11 | 11,824,500 |
| CPR-CloudSat | 1 | F07/T08 | 2.62 | 6.46 | -6.33 | -3.69 | 3.34 | 8.91 | 11.22 | 160,921 |
| DPR-GPM | 1 | F07/T08 | -2.04 | 3.39 | -6.50 | -4.01 | -2.62 | -0.44 | 4.40 | 45,064,500 |
| DPR-GPM | 2 | F07/T08 | -1.46 | 3.76 | -5.71 | -3.66 | -2.43 | -0.23 | 8.73 | 15,381,959 |
| DPR-GPM | 1 | F07/A12 | -22.87 | 3.36 | -27.52 | -24.76 | -23.39 | -21.11 | -16.66 | 9,989,693 |
| DPR-GPM | 2 | F07/A12 | -22.68 | 4.19 | -27.50 | -25.21 | -23.77 | -21.33 | -12.04 | 3,341,397 |

**Table 1.** Statistics of the differences between simulated and observed reflectivities are reported, including the mean, standard deviation (Std), percentiles from the 5th (P5) to the 95th (P95), and the number of observations (nobs). The PSDs for liquid and frozen hydrometeors are also included in the table.

Figure 1 shows simulated versus observed reflectivity values for CloudSat CPR on August 19, 2009, at 17:19:00 UTC. During this time, CloudSat passed over the eye of a tropical cyclone, providing excellent vertical profiling. The input hydrometeor profiles were obtained from a retrieval dataset derived from multiple instruments aboard NASA's A-Train constellation. Simulations and observations are shown only for atmospheric layers above the melting level (273 K), since the simulations do not account for rain effects.

Discrepancies between observed and simulated reflectivities can be attributed to a combination of inaccuracies in the input profiles, forward model errors, and observational biases. Both the simulations and observations show reflectivities reaching up to 30 dBZ. While the results are generally consistent, simulated reflectivities tend to be slightly higher than those observed. In this case, a sector snowflake habit was used to represent snow particles; however, as shown later, using alternative particle shapes can reduce the simulated reflectivities and improve agreement with observations.





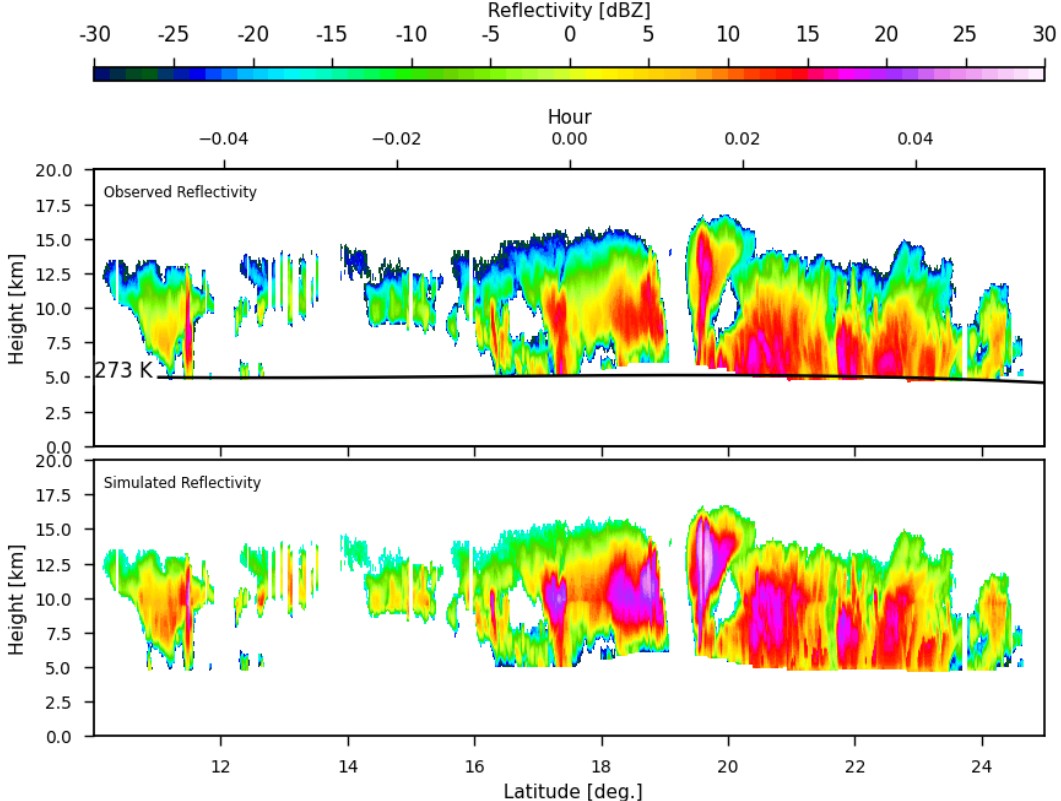

**Figure 1.** Simulated vs. observed CloudSat CPR reflectivities for Hurricane Bill on August 19, 2009, at 17:19 UTC. Simulations include contributions from liquid cloud, ice cloud, and snow water content.

Since no EarthCARE overpasses of tropical cyclones were available during the relatively short period of data availability,
we used simulations from a global dataset to compare with CPR observations from both EarthCARE and CloudSat, as shown in Figure 2. The EarthCARE retrieval dataset discussed in Section 3 does not provide separate water content values for ice and snow particle types. Therefore, the hydrometeors included in the EarthCARE simulations are liquid cloud and snow, while for CloudSat, the simulations incorporate liquid cloud, ice cloud, and snow water content.

Overall, there is good consistency between simulated and observed reflectivity values from both CloudSat and EarthCARE.
On average, EarthCARE CPR shows better agreement with simulations, with a mean difference of 0.59 dBZ compared to 2.62 dBZ for CloudSat. However, when the data are grouped into 30 bins and percentiles are compared, CloudSat shows overall a better agreement. For both instruments, the simulations tend to overestimate reflectivity at lower values but underestimate it at higher reflectivity values.

Figure 3 shows observed versus simulated GPM DPR reflectivities for three different tropical cyclones from 2017. Unlike
CPR, which has a narrow, nadir-viewing beam, the DPR features a wider swath width. As a result, the field of view covering





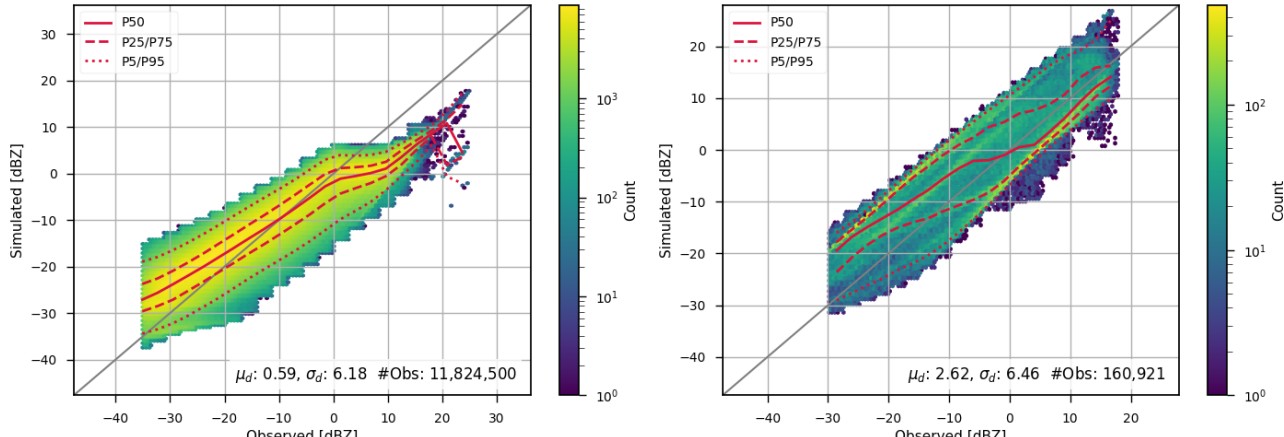

**Figure 2.** CPR simulated reflectivities from both EarthCARE (left) and CloudSat (right) compared with observed reflectivities using a global dataset. Multiple percentiles are also plotted, including the 5th (P5), 25th (P25), 50th (P50), 75th (P75), and 95th (P95). The mean ($\mu_d$) and standard deviation ($\sigma_d$) of the differences, as well as number of observations (#obs) are also displayed on the plots.

the cyclone's eye was not necessarily located at nadir, and in some cases, DPR did not capture observations directly over the storm center. Consequently, the eye of the cyclone may not be visible in these plots.

DPR observations are primarily sensitive to precipitation and liquid cloud water content, and thus reflectivities exceeding the sensitivity threshold are mostly observed below the melting layer. Across all three cyclones, there is strong agreement between observed and simulated reflectivities. For these simulations, we used CRTM cloud coefficients generated using the Thompson particle size distribution (PSD) (Thompson et al., 2004, 2008), and included both liquid cloud and rain water content.

## 4.2 Impact of PSD on Ku/Ka

As shown in Equations 4 and 6, the particle size distributions are used to compute bulk backscattering coefficients from single-particle scattering properties. Over the years, many PSDs have been developed, typically tailored to individual hydrometeor types such as rain or snow. PSDs may be defined using single- or double-moment schemes and often require input parameters such as effective radius or water content. The bulk scattering databses used in fast radiative transfer models are generally parameterized based on the input required by PSD (water content or effective radius), frequency, and temperature.

Figure 4 shows GPM DPR observations compared with simulations conducted using either Abel (Abel and Boutle, 2012) or Thompson (Thompson et al., 2008, 2004) PSDs. Simulations based on the Abel and Boutle (2012) PSD systematically underestimate observed reflectivities, showing a consistent bias of more than 22 dBZ. These simulations rarely exceed 40 dBZ, whereas observations reach up to 60 dBZ. In contrast, simulations using Thompson et al. (2004) PSD show much better agreement with observations, both in magnitude, reaching reflectivities up to 60 dBZ, and in reduced systematic bias.

In the case of simulations based on Thompson et al. (2004) PSD, the agreement is notably strong. In both bands, the 25th and 75th percentiles are very close to the median (50th percentile), showing that half of the data have a difference close to the



**Figure 3.** Observed vs. simulated GPM DPR reflectivities for Ku (left) and Ka (right) bands during Hurricane Irma (September 5, 2017, 16:50 UTC), Hurricane Maria (September 18, 2017, 02:01 UTC), and Hurricane Jose (September 18, 2017, 03:39 UTC).



**Figure 4.** Impact of PSD on simulated GPM DPR reflectivities using a global dataset: comparison between Thompson et al. (2004) PSD (top) and Abel and Boutle (2012) PSD (bottom). The left panels show the KuPR channel, and the right panels show the KaPR channel. The 5th, 25th, 50th, 75th, and 95th percentiles are also shown to indicate how the data are distributed. The mean ($\mu_d$) and standard deviation ($\sigma_d$) of the differences, as well as number of observations (#obs) are also displayed on the plots.

median. It is important to note that these simulations do not account for frozen hydrometeors, which may contribute additional reflectivity in certain cases and could explain some of the residual discrepancies.

## 4.3 Impact of Cloud Type on W-band

This section explores the sensitivity of the EarthCARE W-band to various particle shapes for snow water content. The CRTM includes an extensive scattering database for snow and other frozen hydrometeors (Moradi et al., 2022), which we utilized to
evaluate the impact of different snow hydrometeor shapes on simulated reflectivity. Figure 5 illustrates the sensitivity of CPR




w-band to various shapes used to define scattering based on snow water content. Since attenuation can be influenced by several hydrometeor types as well as water vapor, we focus here on non-attenuated reflectivity to isolate the effects of particle shape.

Figure 5 shows the differences in W-band reflectivity computed using various frozen hydrometeor habits: large plate aggregate (LPA), sector snowflake (SSF), flat three-bullet rosette (F3BR), icon snow (Icon), gem snow (Gem), large column

aggregate (LCA), and six-bullet rosette (6BR). LPA is used as the reference habit, as it generally produces the highest reflectivity values, making it easier to interpret differences relative to it.

Overall, the differences between reflectivities are more pronounced at lower reflectivity values. The smallest differences are seen between LPA and Gem snow, followed by LPA and SSF. For F3BR, the difference compared to LPA is about $-4$ dBZ at lower reflectivities, decreasing to approximately $-2$ dBZ for higher reflectivity values. The difference between LPA and LCA

remains around 4 dBZ, while Icon snow and 6BR differ by approximately 2 dBZ relative to LPA.

Previous studies by Geer and Baordo (2014) and Moradi et al. (2022) have demonstrated that it is possible to identify the optimal particle shape that produces the best agreement with observations for a specific NWP system. While it is not practically feasible to determine the exact hydrometeor shape present in each model grid box, and in reality, multiple shapes may coexist, this analysis can guide the selection of the most compatible particle shape for a given NWP model.

**5 Conclusions**

In this study, we evaluated the performance of a radiative transfer model across multiple radar frequencies using observations from EarthCARE CPR, GPM DPR, and CloudSat CPR. The analysis focused on both the sensitivity of different frequencies to hydrometeor types and the impact of PSDs and hydrometeor shape assumptions on simulated radar reflectivities.

Our results demonstrate that forward simulations using well-characterized PSDs and hydrometeor profiles can reproduce

observed reflectivities with good accuracy, particularly when using retrievals aligned with radar measurements.

We showed that different snow particle habits introduce systematic differences in simulated reflectivities, with the choice of hydrometeor shape affecting the agreement with observed W-band data by several dBZ. These findings support the idea that radiative transfer model accuracy is highly dependent on appropriate microphysical assumptions, particularly for high-frequency radar bands. Similarly, GPM DPR simulations were shown to be highly sensitive to the choice of PSD, with the

Thompson PSD (Thompson et al., 2004, 2008) producing reflectivity magnitudes more consistent with observations than the Abel PSD (Abel and Boutle, 2012).

Future work will include expanding the analysis to cover more hydrometeor types (e.g., hail, graupel), and also determining observation errors for the assimilation of spacebrone radar observations in the NWP models.

*Code and data availability.* The Community Radiative Transfer Model is available from the Joint Center for Satellite Data Assimilation

GitHub repository https://github.com/JCSDA/CRTMv3. EarthCare CPR data can be accessed through the Japan Aerospace Exploration





**Figure 5.** Impact of snow hydrometeor shape on simulated radar reflectivity. Values indicate the difference between simulations using each particle shape and those using the LPA habit.



Agency at https://www.eorc.jaxa.jp/EARTHCARE/index.html. CloudSat CPR data are provided by Colorado State University's dedicated repository https://www.cloudsat.cira.colostate.edu/, and GPM DPR data are available from NASA at https://disc.gsfc.nasa.gov/.

*Author contributions.* Isaac Moradi: Conceptualization, Methodology, Formal analysis, Investigation, Software, Visualization, Writing – original draft, Writing – review and editing. Satya Kalluri: Conceptualization, Methodology, Formal analysis, Writing – review and editing. Yanqiu Zhu: Conceptualization, Methodology, Formal analysis, Writing – review and editing.

*Competing interests.* The authors declare that they have no conflict of interest.

*Acknowledgements.* This study was supported by NASA grant 80NSSC21K1361 and NOAA grant NA24NESX432C0001 (Cooperative Institute for Satellite Earth System Studies - CISESS) at the University of Maryland/ESSIC.




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
