# Peer review of "Forward Modeling of Spaceborne Active Radar Observations"

_EGUsphere, 2025_

## Author Comment (AC1)

**Reviewer I**

The work is well written and presented. The validation of the CRTM radar operator along with an exploration of its sensitivities to microphysical assumptions is significant and of interest to the community. The CRTM radar model, being supported, open source and publicly available, is likely to be used in many future studies as well as in weather models. However, the study is currently incomplete because it is lacking any comparison between CRTM and the radar models being used by the level 2 data providers. This is the first place to look to explain the differences between the CRTM simulations and the observations. There are a number of other smaller issues to be addressed as well.

We would like to express our sincere thanks to Dr. Geer for his positive remarks and thoughtful suggestions. We fully agree that our findings are influenced by the retrievals, particularly by the radiative transfer code and its underlying assumptions. In the revised manuscript, we now provide a clearer discussion comparing the microphysical assumptions used in our study with those applied in the forward modeling components of the retrieval system.

**Main points**

The methodology of this study is slightly circular, as explained above. Although this approach is justifiable for a model validation study, it needs to be more clearly flagged to readers, and its implications followed up in more detail. In this context, if the the CRTM radar operator was identical to the forward model used in the level 2 retrievals, and if no information had been lost, then we would hope for an almost exact match between the CRTM simulations and observations. Already, the level of agreement between simulations and observations in terms of spatial structures is nearly exact, for example in Figure 1, which clearly shows that despite other passive data being used in the CloudSat retrievals, they must be very much dominated by the reflectivity.

We thank the reviewer for this insightful comment. We have added two paragraphs near the beginning of Section III to address this issue. While we agree that some dependencies exist, we still believe, as noted by the reviewer, that these retrieval products provide a unique opportunity for evaluating forward models.

Despite the close spatial agreement between simulations and observations, there are systematic differences in the reflectivities which are of interest and are most likely explained by differing microphysical assumptions in the two forward models, or by different model methodology (such as treatments for single or multiple scattering, or cloud overlap and inhomogeneity within the beam). Another possible explanation is if some of the reflectivity-generating hydrometeor mass has been lost between the retrieval and the input to the simulation (for example, if the snow mass has been somehow discarded in the DPR comparison, which only uses cloud and rain hydrometeor profiles). I would hope that all results could be explored in more detail in this framework, following these suggestions:

The discrepancies are likely due to a combination of differences in MP assumptions and the exclusion of some hydrometeor types. We have expanded our explanation of these points in the revised manuscript. As the reviewer also noted in later comments, if a hydrometeor type is not explicitly retrieved, it might be implicitly included in the water content profiles for other habits. We have also include two tables: one outlining the MP assumptions used in the retrievals and another summarizing the water content profiles and associated shapes employed in the CRTM simulations.

1) At a high level (e.g. abstract, introduction, conclusions) the work should indicate more clearly that the source of the hydrometeor profiles is the observations themselves, and that the study can be more clearly seen as a consistency validation against the existing radar models employed in the level 2 retrievals.

We thank the reviewer for this comment. We have discussed this in detail in Section III and also provided a brief explanation in the Introduction as follows.

> There are three main sources of input profiles for validating RT models: in situ measurements, NWP simulations, and retrievals from combined active–passive sensors. In this study, we use retrieval products to avoid temporal and spatial mismatches between the observations and input profiles. While these retrievals involve radiative transfer assumptions, they still provide a practical and reliable basis for evaluating forward models.

2) The work needs to detail more exactly the radar forward modelling being used by the data providers, especially the basic assumptions such as single particle scattering model or habit and particle size distribution, and any more advanced methods used for multiple scattering, cloud overlap or inhomogeneity. Also a little more information needs to be provided on the CRTM configuration in these areas. I imagine it should be possible to provide a new table comparing exactly the configuration of CRTM to the configurations of the forward models used in the Cloudsat, EarthCARE and DPR retrievals.

We've added a new table (Table 1) that compares the assumptions used in CRTM with those in the radiative-transfer models behind the retrievals. Where appropriate, we also updated the manuscript to reflect the information summarized in Table 1. In addition, we have included a second table that lists the water-content profiles used as input to CRTM.

3) The work needs to more clearly summarise the sources of hydrometeor mass being used in each comparison, since it differs. Again, this would be very helpful if presented in tabular form. It is also important to double check the retrieval methods and see if any hydrometeor mass has been missed. For the DPR example, where CRTM is only supplied with liquid water and rain profiles, if all the reflectivity above the melting layer has been attributed to supercooled droplets in the DPR retrieval, then even if in practice some of this reflectivity has been generated by frozen particles, it would not matter for the consistency validation, as long as the rain and cloud scattering models were identical. But if the DPR retrieval represents frozen particles separately, then mass may have gone missing in the comparison. Given the relatively good agreement shown in Figure 3, there unlikely to be missing mass in the DPR comparison, but it would be important to be sure, and this would help reduce the number of knowledge gaps in the study.

We have included a new table showing the water-content profiles used as input to CRTM. In addition, we believe that the mass associated with excluded hydrometeor habits is not entirely missing but is implicitly represented by other habits. For instance, graupel water content may be absent from the retrievals, but it is likely incorporated into the ice or snow water-content categories.

4) Based on the comparisons listed at points 2 and 3 above, it should be possible to explain many of the systematic differences between simulations and observations. For example, the conclusion that the Thompson PSD worked better the Abel PSD in simulating DPR is not useful on its own (this is not an independent validation) but it is useful when compared to the assumptions used by the DPR retrieval team.

We have better explained the results but we believe the error due to Abel's model (more than 20 dBZ) is more than anything that can be explainable by differences in PSD distribution. This is now better discussed in "Section 4.2: Impact of PSD on Ku/Ka".

5) Since there is a large body of existing work on radar simulators, much of which has fed into the forward models used for the Cloudsat, EarthCARE and DPR retrievals, it would be good to acknowledge this and cover it in the introduction too.

We thank the reviewer for the suggestion. We have included a new paragraph in "Introduction" acknowledging the role of previous studies in radar scattering calculations.

**Minor points**

1) Line 15 - radiative transfer models also critically include the surface, not just the atmosphere

The introduction has been substantially revised, and although this specific sentence no longer exists in the revised version, the reviewer's point has been taken into account.

2) Line 35 referencing the ZmVar model: there is a more recent publication that could be useful to reference here: ''Direct 4D-Var assimilation of space-borne cloud radar reflectivity and lidar backscatter. Part I: Observation operator and implementation, by MD Fielding, M Janisková - Quarterly Journal of the Royal Meteorological Society, 2020''. Here, especially the two-column treatment of cloud overlap for the attenuation calculation is quite novel.

We have included a new paragraph in "Introduction" discussing the new publication by Fiedling and Janisková (2020).

3) Line 72, the radar equation needs a little more explanation, since this refers more precisely to the equivalent radar reflectivity (or reflectivity factor) rather than the actual reflectivity. Here there should also be some support from prior textbooks or papers: for example Grant Petty's book has a good introduction of these concepts.

We have completely revised the section describing the radar forward model. Please see the revised Section 2.2.

4) Line 225: ''Discrepancies between observed and simulated reflectivities can be attributed to a combination of inaccuracies in the input profiles, forward model errors, and observational biases''. Based on discussions above, I would not expect observational biases to have any relevance here.

We thank the reviewer and have revised the text as follows:

> Several factors contribute to discrepancies between observed and simulated reflectivities. These include differences in PSDs and particle shape assumptions used in the CRTM calculations compared with the assumption made during retrieving these products from observations, uncertainties in the input profiles, and the exclusion of certain hydrometeor types from the simulations.

5) Line 236, ''CloudSat shows overall a better agreement. For both instruments, the simulations tend to overestimate reflectivity at lower values but underestimate it at higher reflectivity values.'' This discussion needs to be more carefully framed because the CloudSat

results are within tropical cyclones whereas the EarthCARE results are global. The difference in agreement could be purely due to the different datasets and have nothing to do with the retrievals or the sensors themselves.

Both CloudSat and EarthCare results are based on global datasets, and this has been reflected in both the text and the figure caption. This section has also been largely revised.

6) Figure 2a is hard to reconcile with Fig 1, which shows simulations in convective cores reaching 30 dBZ, compared to observations rarely exceeding 20 dBZ. Figure 2a shows the opposite and the text concludes that the simulations are too low, not too high.

Figure 2a corresponds to the EarthCARE CPR, whereas Figure 1 shows CloudSat CPR data. The appropriate comparison is therefore between Figure 2b and Figure 1. However, Figure 2b uses global data, while Figure 1 is based on a tropical-cyclone overpass. The section discussing the results has been substantially revised in the updated manuscript.

7) E.g. line 246 "included liquid and rain content". Partly repeating earlier main points, it would have been much easier to follow the results if the basic details of the CRTM simulations had been presented and summarised, ideally in a table. This would include basic details like particle shape and PSD choices for each hydrometeor type in each simulation, along with the hydrometeor types being used in each case.

We have included a new table that presents the CRTM assumptions and the water content profiles used in each set of simulations.

8) Figure 3 and others show a freezing level on the observations but not simulations, which makes it harder to compare the two. To make things more visually consistent and comparable, the line should ideally be on both panels.

We have included the freezing levels at both panels.

9) Line 267 "we focus on non-attenuated reflectivity". This is a bit confusing as, as far as I could see, the text has not yet clearly stated whether any of the results are based on attenuated or non-attenuated reflectivity. This needs a clearer signposting earlier in the text. I also don't like the term "non-attenuated reflectivity" relating to observations, because what these actually are is "attenuation corrected reflectivity".

We agree with the reviewer and have changed it to attenuation corrected reflectivity. We have also added the following statement at the beginning of the "Results" section to emphasize that the results are generally based on attenuated reflectivity: "The results presented here are based on attenuated reflectivity, unless noted otherwise."

10) Figure 4: the difference in the x-axis ranges makes comparison very hard. Please ensure consistency across all the panels, if possible.

We have revised the figure so that all plots use the same range.

11) Line 267 "do not account for frozen hydrometeors". This is picking up on a point made earlier, that the more important question is whether the retrievals attempt to represent frozen hydrometeors separately, or whether any reflectivity above the freezing level is assumed to be explained by supercooled water droplets (assuming also that melting particles are not separately represented either).

We have revised the Results to better explain the exclusion of certain hydrometeors from the simulations. We believe these habits are not being completely excluded, but are instead implicitly included within the retrieved water-content categories. For instance, if both supercooled water and frozen habits were truly excluded, then in Figure 3 we would not see any simulated reflectivity above the melting layer.

**Technical and grammar**

1) The second part of the final sentence in the abstract seems to convey no concrete meaning - please rephrase for clarity or remove: ''underscoring their potential to improve the assimilation of spaceborne radar data in NWP models''. Exactly what has the potential to improve assimilation is not clear.

The abstract has been revised for clarity. The text now reads as:

> The sensitivity of forward radar simulations to microphysical assumptions, underscores their importance for the assimilation of radar observations in numerical weather forecast models.

2) Line 22 CRTM is a ''pivotal collaborative model''. This doesn't seem to convey much scientific meaning and should be removed or explained more clearly.

Thank you for noting this. We have completely revised the introduction for better consistency.

3) line 28: space radars ''provide vertical profiles of cloud and precipitation structures'' - this language is too loose. They provide reflectivity profiles giving information on vertical structures of cloud and precipitation.

We have fully revised the introduction for consistency and clarity of language.

4) line 50: ''supports an operational environment'' does not seem to convey a clear meaning and could be removed or rephrased.

Thank you for carefully reading the manuscript. As noted, we have fully revised the introduction for consistency and clarity of language.

5) equation 1: please use a more self-consistent mathematical notation both here and throughout the paper, and most critically of all, please explain what it is. Here the vector y is bold and the vector x is non-bold italic, for example. The use of bold capitals for both the observation operator H (presumably a nonlinear function) and the background error covariance B (a matrix) is potentially confusing. Typically bold capital H is used for the Jacobian matrix of the nonlinear operator H, which itself is usually written in non-bold italic capitalised or similar.

Thank you for noting the discrepancy. We have revised the text to prevent any confusion in the interpretation of the equations.

6) equation 4 and others: please explain or remove the comma notation (for multiplication?) which does not seem to be very standard.

We have revised all of these equations, as the commas were mistakenly added due to a LaTeX command that did not work correctly.

7) Line 89 V(D) is not strictly the particle volume, in the case of non-spherical particles.

The reviewer is correct, and the text has been updated as follows:

> where $m(D)$ is the particle mass (kg), $\rho(D) = m(D)/V(D)$ is the particle density, and $V(D)$ is the equivalent spherical volume. For convenience, $V(D) = \pi D^3/6$ is used to represent the volume of a sphere with diameter $D$, even though the actual particles may be nonspherical. This approximation provides a consistent way to relate particle mass and size through an equivalent-volume definition.

8) line 240: ''as a result ...   DPR did not capture observations directly over the storm centre'' is incorrect. The swath of DPR made it more likely to capture the storm centres (as opposed to nadir viewing sensors like CloudSat). In any case this is a separate message and requires a separate sentence for clarity.

What this really means is that the cyclone's eye appears missing because the storm center was not located in the middle of the scan, and a tilted scan may not clearly capture the eye. In some scans, the storm was entirely absent as well.

---

## Author Comment (AC2)

**Reviewer II**

The paper "Forward Modeling of Spaceborne Active Radar Observations" presents the implementation of a forward operator for spaceborne radar reflectivity observations within the CRTM model. The operator is evaluated for three types of spaceborne observing systems covering a broad frequency range (Ku, Ka, and W). The overall objective of the study is appealing: providing a spaceborne reflectivity simulator within an operational radiative transfer framework is a valuable step toward future data assimilation applications.

We sincerely thank the reviewer for taking the time to review our manuscript and for providing valuable comments.

The paper is well written and easy to follow. However, I think the order of the content in Section 2 could be re-arranged and extended with more details (see below). Besides, contrary to the approach used in other papers about the validation of spaceborne reflectivities, to avoid uncertainties in the prediction of hydrometeors using NWP models, the authors preferred to validate the simulator using hydrometeor profiles that have been retrieved using observations. I think it is a nice approach. However, details about the retrievals are not explained in the current manuscript and should be added in a future version of the manuscript.

We have revised Section II following the reviewer's suggestions and expanded the discussion of the retrieval products to provide additional clarity.

**Abstract:**

L4: "active radar module": I think "active" can be removed as a radar is always an active sensor. (same for the title)

Thank you for the suggestion. We have removed the word "active" accordingly.

**Introduction:**

L26 to L30: I recommend the authors to add the appropriate references on the current existing spaceborne missions. PMR onboard FY3G is not mentioned as a current spaceborne mission (see for instance: https://rmets.onlinelibrary.wiley.com/doi/10.1002/qj.4964). The authors could also add future planned spaceborne radar observing systems (e.g. WIVERN, INCUS).

We have revised the paragraph and added the additional current and planned spaceborne missions as suggested. Please see Lines 26-31 in the updated manuscript.

L36: The authors could also cite the work of Fielding et al. (2021).

This citation has now been discussed.

**Section 2 :**

« forward model in DA » here is a misleading title as the forward operator is then never applied to any NWP model in this paper, and there are no DA experiments in this paper. Therefore, I think the introduction of section 2 is out of the scope of the current paper (especially with the general equation of the 3DVar).

We believe that, since the ultimate goal of this work is the assimilation of radar observations, it makes sense to keep this introduction, especially given that the first reviewer specifically suggested relating our formulation to the data assimilation concepts.

‘‘Radar forward model'' would better fit into after the CRTM section as the radar solver is included in CRTM

We have re-arranged the section according to the reviewer's recommendation.

Section 2.1: it is not written how the authors account for the subgrid variability of hydrometeors, especially if the final goal is to apply this operator to global NWP models. Also, the authors don't mention the dielectric properties, which are very important to compute scattering properties.

Given that we have used the retrieved products for the assessment in this work, we did not account for beam filling. In our ongoing data assimilation work, we are computing a beam-filling index using the variability among the profiles within each model grid box. However, this effort is still in progress and is not included in the current manuscript.
We have, however, incorporated a description of how the dielectric properties were computed in the revised manuscript.

**Section 3:**

L115: it would be informative to add the radar frequencies of AOS

We have removed that short paragraph because the future of AOS is unclear and we are not really sure what frequencies will be selected for the AOS.

L122-124: I personally disagree with the authors. Global dataset of hydrometeors exists with NWP models, and have been used in many studies to validate radar forward operators (Di Michele et al. 2012; Fielding et al. 2021; Ikuta et al. 2016, David et al. 2025, etc. . . )

We have revised Section III and added further clarification to ensure that the reviewer's point is fully taken into account.

L123: The authors should explain the approximations made in the retrieval as this is crucial to explain the results of the sensitivity study to the PSD, and to the particle shapes (section 4). For example, is there any assumption on the PSD in the retrieval; which could then explain why the results are closer if the Thomson PSD is used in CRTM (Fig 4)?

We have revised the manuscript and provided additional information on the retrieval assumptions to clarify this point.

L134-136: In my opinion the sentence is a bit too long.

We have split the sentence into two for improved clarity, as follows:

> Specifically, we utilized the updated vBb/v1.1 version of the algorithm, which provides cloud masks and cloud type classifications. This version also improves detection near the ground by reducing surface clutter contamination compared to previous versions.

 The authors should add a reference when they argue that the graupel signal is similar than the one of ice, or to demonstrate this point. Indeed, this sentence is a bit counterintuitive as the properties of both species are different in the nature (mass, PSDs, dielectric properties).

We have clarified this point by referring to the results of Moradi et al. (2023), whose Figure 8 shows that assigning the same water content to ice or graupel leads to only small differences in reflectivity, with differences approaching 5 dBZ only in strongly convective regions.

> Moradi et al. (2023) demonstrate that fully swapping the ice water content between ice clouds and graupel can introduce differences of up to 5 dBZ in the heavily convective regions of tropical cyclones, although in most areas the differences are substantially smaller.

**Section 4:**

The authors should explain why the sensitivity study is not performed at the three frequencies (for the PSDs as well as on the shapes).

This was limited by our scattering database used in CRTM. We have included the following explanation in the text:

> However, the CRTM cloud scattering database includes only spherical particles for rain; therefore, we have limited this sensitivity study to the W-band, as DPR frequencies are primarily sensitive to rain.

L193: Another point is that there is usually a spatial and temporal mismatch between NWP models and radar observations, which makes difficult to disentangle errors in the forward operator from spatial and temporal mismatch (see for instance section 4 in Borderies et al. 2018)

We have edited this paragraph as follows to be consistent with the previous statement we made in Section III:

> As noted earlier, there are several sources that can be used to validate a forward model. For this study, we have chosen retrieval products to avoid temporal and spatial mismatches with the observations. While we acknowledge that some dependence on radiative transfer exists in these retrievals, CRTM is not used during the retrieval process. Therefore, we consider these retrieval products a reliable source for validating the RT model.

L197-200: the authors should add the end of the period of study.

We believe the current text already indicates the study period by specifying that it covers the first week of May.

In the text, the units of the comparisons of Table 1 should be in dB, and not dBZ (a difference between two dBZ is in dB).

We have corrected the text and updated the corresponding figure to use the appropriate unit, i.e., dB.

Legend of Table 1: what are the shapes used for the simulations?

We used sector snowflakes to represent snow particles. This information has been added in a table and the text is revised to better reflect this.

L221: Is there any reference for the retrieval?

We have added a few new references for the retrievals.

L222: The authors should explain why the simulations do not account for rain effect

We have explained the reason for excluding the rain water content from the simulations as follows:

> The rain backscatter was excluded because no reliable rain water content was available that was collocated with the CloudSat CPR observations. Because the CPR 94-GHz radar is primarily sensitive to frozen hydrometeors, excluding rain water content while not ideal, is unlikely to have a substantial impact on the results.

L260: the authors should explain why the simulations do not account for frozen hydrometeors. It is confusing for the reader.

This has been clarified in the revised version as follows:

> The retrieval database used to prepare the input water content profiles required by CRTM did not include water content for frozen hydrometeors such as snow and ice. As a result, these simulations do not account for frozen hydrometeors, which may contribute additional reflectivity in certain cases and could explain some of the residual discrepancies. However, because DPR frequencies are primarily sensitive to rain and liquid cloud, excluding frozen hydrometeor water content, while potentially affecting specific situations such as heavy snow, is unlikely to impact the overall results.

L266:''w-band''-> ''W-band''

Thank you! This has been corrected to ''W-band.''

L267: in that case, the authors should say that they use the corrected reflectivity in the observations.

It is expected that the results for attenuation-corrected reflectivity and attenuated reflectivity would be similar, because the shape of snow particles has little impact on attenuation, which is primarily sensitive to rain, liquid cloud, and species such as water vapor.

I think that this result about the larger differences for smaller content is very interesting. Is there any paper in the literature to support this result, or any physical argument to support these findings? Is this larger difference at smaller contents only due to the shape, or also to the mass-diameter relationships which is associated to each shape? I would recommend the authors to add some more explanations about this point.

We believe this is largely due to the logarithmic scale used in computing reflectivity in dBZ. When these differences are converted to physical units, such as $\text{mm}^6/\text{m}^3$, the absolute differences become much larger at higher reflectivity values. For instance, at $-30$ dBZ with a $4$ dBZ difference, the difference in linear units is approximately $1.51 \times 10^{-3}$ $\text{mm}^6/\text{m}^3$, whereas at $30$ dBZ with a $1$ dBZ difference, it is approximately $2.59 \times 10^2$ $\text{mm}^6/\text{m}^3$. The text has also

be updated to reflect this.

L290: In my opinion the first perspective would be to test this forward operator on NWP model fields (at least before estimating any observation errors for DA).

Thank you for your suggestion. This work is already underway, and the new forward operator is currently being tested within both the NASA and NOAA NWP systems.

---

## Author Response (AR2)

We thank the reviewer for the careful reading of the revised manuscript and for the constructive comments. Our responses to the remaining comments are provided below.

1. The sentence in the abstract has been revised as follows:

For the GPM DPR, reflectivities simulated using the Thompson PSD showed closer agreement with the observations than those using the Abel PSD; this agreement should be interpreted in the context of the limited independence between the observations and the retrievals used as input to the CRTM, which themselves rely on PSD-related assumptions.

2. The sentence in the conclusions has been clarified to emphasize the dependence of the retrievals on PSD-related assumptions.

Similarly, GPM DPR simulations were shown to be highly sensitive to the choice of PSD, with the Thompson PSD \citep{thompson2004,thompson_explicit_2008} producing reflectivity magnitudes more consistent with observations than the Abel PSD \citep{abel2012}. However, this apparent consistency should be interpreted cautiously, since the retrievals used as input to the CRTM are not fully independent of the observations and themselves rely on underlying PSD assumptions. As a result, these findings may reflect greater consistency with the assumptions employed during the retrieval process rather than definitive evidence of deficiencies in any particular PSD, which would require independent evaluation using, for instance, in situ measurements.